# Basic Binary Convolution Unit for Binarized Image Restoration Network

**Bin Xia**[1], **Yulun Zhang**[2], **Yitong Wang**[3], **Yapeng Tian**[4],
**Wenming Yang**[1]*, **Radu Timofte**[5], **and Luc Van Gool**[2]
[1]Tsinghua University  [2]ETH Zürich  [3]ByteDance Inc
[4]University of Texas at Dallas  [5]University of Würzburg

## Abstract

Lighter and faster image restoration (IR) models are crucial for the deployment on resource-limited devices. Binary neural network (BNN), one of the most promising model compression methods, can dramatically reduce the computations and parameters of full-precision convolutional neural networks (CNN). However, there are different properties between BNN and full-precision CNN, and we can hardly use the experience of designing CNN to develop BNN. In this study, we reconsider components in binary convolution, such as residual connection, BatchNorm, activation function, and structure, for IR tasks. We conduct systematic analyses to explain each component's role in binary convolution and discuss the pitfalls. Specifically, we find that residual connection can reduce the information loss caused by binarization; BatchNorm can solve the value range gap between residual connection and binary convolution; The position of the activation function dramatically affects the performance of BNN. Based on our findings and analyses, we design a simple yet efficient basic binary convolution unit (BBCU). Furthermore, we divide IR networks into four parts and specially design variants of BBCU for each part to explore the benefit of binarizing these parts. We conduct experiments on different IR tasks, and our BBCU significantly outperforms other BNNs and lightweight models, which shows that BBCU can serve as a basic unit for binarized IR networks. The code is available at https://github.com/Zj-BinXia/BBCU.

## 1 Introduction

Image restoration (IR) aims to restore a high-quality (HQ) image from its low-quality (LQ) counterpart corrupted by various degradation factors. Typical IR tasks include image denoising, super-resolution (SR), and compression artifacts reduction. Due to its ill-posed nature and high practical values, image restoration is an active yet challenging research topic in computer vision. Recently, the deep convolutional neural network (CNN) has achieved excellent performance by learning a mapping from LQ to HQ image patches for image restoration (Chen & Pock, 2016; Zhang et al., 2018a; Tai et al., 2017; Xia et al., 2023). However, most IR tasks require dense pixel prediction and the powerful performance of CNN-based models usually relies on increasing model size and computational complexity. That requires extensive computing and memory resources. While, most hand-held devices and small drones are not equipped with GPUs and enough memory to store and run the computationally expensive CNN models. Thus, it is quite essential to largely reduce its computation and memory cost while preserving model performance to promote IR models.

Binary neural network (Courbariaux et al., 2016) (BNN, also known as 1-bit CNN) has been recognized as one of the most promising neural network compression methods (He et al., 2017; Jacob et al., 2018; Zoph & Le, 2016) for deploying models onto resource-limited devices. BNN could achieve $32\times$ memory compression ratio and up to $64\times$ computational reductions on specially designed processors (Rastegari et al., 2016). Nowadays, the researches of BNN mainly concentrate on high-level tasks, especially classification (Liu et al., 2018; 2020), but do not fully explored in low-level vision, like image denoising. Considering the great significance of BNN for the deployment of IR deep networks and the difference between high-level and low-level vision tasks, there is an

---

*Corresponding Author: Wenming Yang, yang.wenming@sz.tsinghua.edu.cn

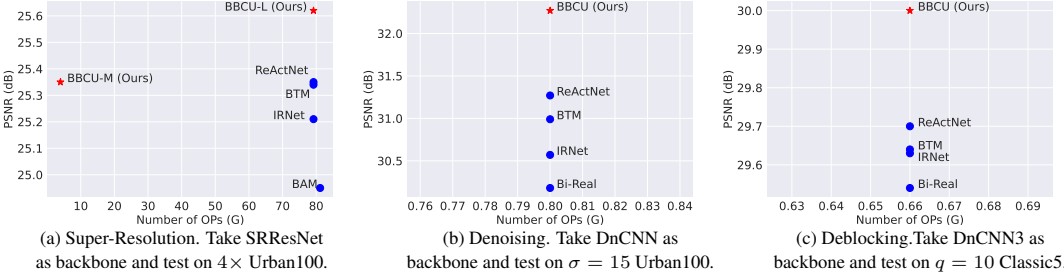

Figure 1: Our BBCU achieves the SOTA performance on IR tasks with efficient computation.

urgent need to explore the property of BNN on low-level vision tasks and provide a simple, strong, universal, and extensible baseline for latter researches and deployment.

Recently, there have been several works exploring the application of BNN on image SR networks. Specifically, Ma *et al.* (Ma et al., 2019) tried to binarize the convolution kernel weight to decrease the SR network model size. But, the computational complexity is still high due to the preservation of full-precision activations. Then BAM (Xin et al., 2020) adopted the bit accumulation mechanism to approximate the full-precision convolution for the SR network. Zhang *et al.* (Zhang et al., 2021b) designed a compact uniform prior to constrain the convolution weight into a highly narrow range centered at zero for better optimization. BTM (Jiang et al., 2021) further introduced knowledge distillation (Hinton et al., 2015) to boost the performance of binarized SR networks.

However, the above-mentioned binarized SR networks can hardly achieve the potential of BNN. In this work, we explore the properties of three key components of BNN, including residual connection (He et al., 2016), BatchNorm (BN) (Ioffe & Szegedy, 2015), and activation function (Glorot et al., 2011) and design a strong basic binary Conv unit (BBCU) based on our analyses.

**(1)** For the IR tasks, we observe that residual connection is quite important for binarized IR networks. That is because that BNN will binarize the input full-precision activations to 1 or -1 before binary convolution (BC). It means that BNN would lose a large amount of information about the value range of activations. By adding the full-precision residual connection for each binary convolution (BC), BNN can reduce the effect of value range information loss.

**(2)** Then, we explore the BN for BBCU. BNN methods (Liu et al., 2020) for classification always adopt a BN in BC. However, in IR tasks, EDSR (Lim et al., 2017) has demonstrated that BN is harmful to SR performance. We find that BN in BNN for IR tasks is useful and can be used to balance the value range of residual connection and BC. Specifically, as shown in Fig. 2 (b), the values of the full-precision residual connection are mostly in the range of - 1 to 1 because the value range of input images is around 0 to 1 or -1 to 1, while the values of the BC without BN are large and ranges from -15 to 15 for the bit counting operation (Fig. **??**). The value range of BC is larger than that of the residual connection, which covers the preserved full-precision image information in the residual connection and limits the performance of BNN. In Fig. 2 (a), the BN in BNN for image restoration can realize the value range alignment of the residual connection and BC.

**(3)** Based on these findings, to remove BN, we propose a residual alignment (RA) scheme by multiplying the input image by an amplification factor $k$ to increase the value range of the residual connection rather than using BN to narrow the value range of the BC (Fig. 2 (c)). Using this scheme can improve the performance of binarized IR networks and simplify the BNN structure (Sec. 4.5).

**(4)** In Fig. 2 (d), different from BNNs (Liu et al., 2020; 2018) for classification, we further move the activation function into the residual connection and can improve performance (Sec. 4.5). That is because activation function would narrow the negative value ranges of residual connection. Its information would be covered by the next BC with large negative value ranges (Fig. 2 (c)).

**(5)** Furthermore, we divide IR networks into four parts: head, body, upsampling, and tail (Fig. 3 (a)). These four parts have different input and output channel numbers. Previous binarized SR networks (Xin et al., 2020; Jiang et al., 2021) merely binarize the body part. However, upsampling part accounts for 52.3% total calculations and needs to be binarized. Besides, the binarized head and tail parts are also worth exploring. Thus, we design different variants of the BBCU to binarize these four parts (Fig. 3 (b)). Overall, our contributions can be mainly summarized in threefold:

- We believe our work is timely. The high computational and memory cost of IR networks hinder their application on resource-limited devices. BNN, as one of the most promising compression methods, can help IR networks to solve this dilemma. Since the BNN-based networks have different properties from full-precision CNN networks, we reconsider, analyze, and visualize some essential components of BNN to explore their functions.

- According to our findings and analyses on BNN, we specially develop a simple, strong, universal, and extensible basic binary Conv unit (BBCU) for IR networks. Furthermore, we develop variants of BBCU and adapt it to different parts of IR networks.

- Extensive experiments on different IR tasks show that BBCU can outperform the SOTA BNN methods (Fig. 1). BBCU can serve as a strong basic binary convolution unit for future binarized IR networks, which is meaningful to academic research and industry.

## 2 RELATED WORK

### 2.1 IMAGE RESTORATION

As pioneer works, SRCNN (Dong et al., 2015b), DnCNN (Zhang et al., 2017), and ARCNN (Dong et al., 2015a) use the compact networks for image super-resolution, image denoising, and compression artifacts reduction, respectively. Since then, researchers had carried out its study with different perspectives and obtained more elaborate neural network architecture designs and learning strategies, such as residual block (Kim et al., 2016; Zhang et al., 2021a; Cavigelli et al., 2017), dense block (Zhang et al., 2018c; 2020; Wang et al., 2018), attention mechanism (Zhang et al., 2018b; Xia et al., 2022a; Dai et al., 2019), GAN (Gulrajani et al., 2017; Wang et al., 2018), and others (Wei et al., 2021; Peng et al., 2019; Jia et al., 2019; Fu et al., 2019; Kim et al., 2019; Fu et al., 2021; Soh et al., 2020; Xia et al., 2022d;c;b), to improve model representation ability. However, these IR networks require high computational and memory costs, which hinders practical application on edge devices. To address this issue, we explore and design a BBCU for binarized IR networks.

### 2.2 BINARY NEURAL NETWORKS

Binary neural network (BNN) is the most extreme form of model quantization as it quantizes convolution weights and activations to only 1 bit, achieving great speed-up compared with its full-precision counterpart. As a pioneer work, BNN (Courbariaux et al., 2016) directly applied binarization to a full-precision model with a pre-defined binarization function. Afterward, XNOR-Net (Rastegari et al., 2016) adopted a gain term to compensate for lost information to improve the performance of BNN (Courbariaux et al., 2016). After that, Bi-Real (Liu et al., 2018) introduced residual connection to preserve full-precision information in forward propagation. IRNet (Qin et al., 2020) developed a scheme to retain the information in forward and backward propagations. Recently, ReActNet (Liu et al., 2020) proposed generalized activation functions to learn more representative features. For super-resolution task, Ma et al. (2019) binarized convolution weights to save model size. However, it can hardly speed up inference enough for retaining full-precision activations. Then, Xin et al. (2020) further binarized activations and used the bit accumulation mechanism to approximate the full-precision convolution for SR networks. Besides, Zhang et al. (2021b) introduced a compact uniform prior for better optimization. Subsequently, Jiang et al. (2021) designed a binary training mechanism by adjusting the feature distribution. In this paper, we reconsider the function of each basic component in BC and develop a strong, simple, and efficient BBCU.

## 3 PROPOSED METHOD

### 3.1 BASIC BINARY CONV UNIT DESIGN

As shown in Fig. 2(a), we first construct the BBCU-V1. Specifically, the full-precision convolution $\mathcal{X}_\mathbf{j}^\mathbf{f} \otimes \mathcal{W}_\mathbf{j}^\mathbf{f}$ ($\mathcal{X}_\mathbf{j}^\mathbf{f}$, $\mathcal{W}_\mathbf{j}^\mathbf{f}$, and $\otimes$ are full-precision activations, weights, and Conv respectively) is approximated by the binary convolution $\mathcal{X}_\mathbf{j}^\mathbf{b} \otimes \mathcal{W}_\mathbf{j}^\mathbf{b}$. For binary convolution, both weights and activations are binarized to -1 and +1. Efficient bitwise XNOR and bit counting operations can replace computationally heavy floating-point matrix multiplication, which can be defined as:

$$\mathcal{X}_\mathbf{j}^\mathbf{b} \otimes \mathcal{W}_\mathbf{j}^\mathbf{b} = \text{bitcount}\left(\text{XNOR}\left(\mathcal{X}_\mathbf{j}^\mathbf{b}, \mathcal{W}_\mathbf{j}^\mathbf{b}\right)\right), \tag{1}$$

$$x_{i,j}^b = \text{Sign}\left(x_{i,j}^f\right) = \begin{cases} +1, \text{ if } x_{i,j}^f > \alpha_{i,j} \\ -1, \text{ if } x_{i,j}^f \le \alpha_{i,j} \end{cases}, x_{i,j}^f \in \mathcal{X}_j^f, x_{i,j}^b \in \mathcal{X}_j^b, i \in [0, C), \tag{2}$$

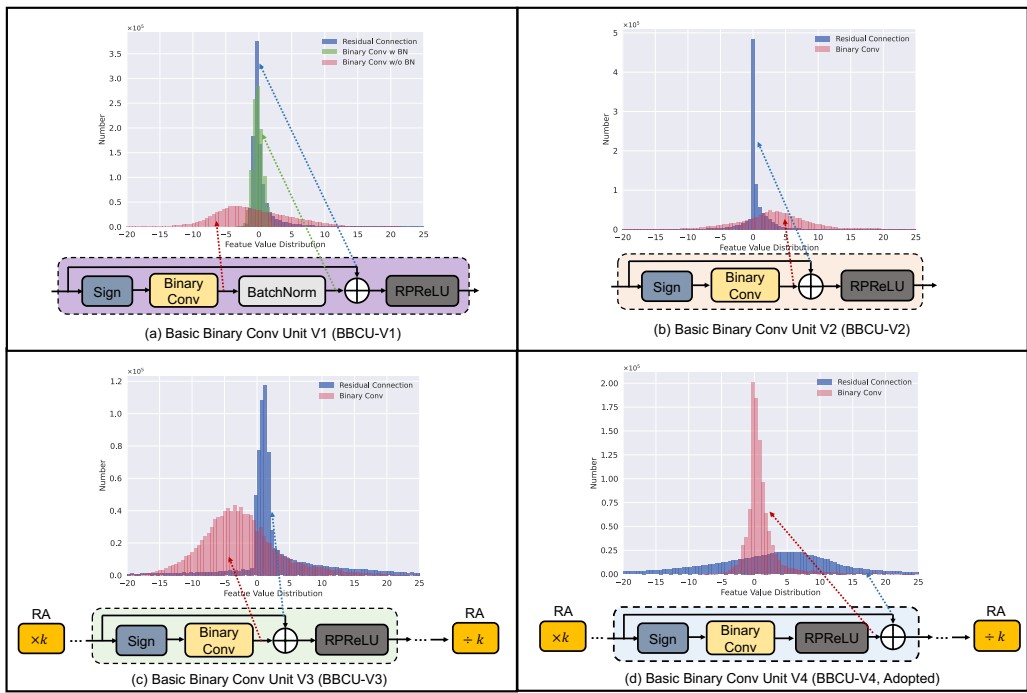

Figure 2: The illustration of the improvement process of our BBCU. (a) The initial BBCU design. (b) We remove BatchNorm to explore its actual function in IR tasks. We find that BatchNorm is essential because it can balance the value range gap between residual connection and binary convolution. (c) We further propose the residual alignment (RA) by multiplying an amplification factor $k$ on the input image to address the value range gap. (d) Based on BBCU-V3, we move the activation function into the residual connection to reduce the negative value information loss.

$$w_{i,j}^b = \frac{\left\|\mathcal{W}_{\mathbf{j}}^{\mathbf{f}}\right\|_1}{n} \mathrm{Sign}\left(w_{i,j}^f\right) = \begin{cases} +\frac{\left\|\mathcal{W}_{\mathbf{j}}^{\mathbf{f}}\right\|_1}{n}, \text{ if } w_{i,j}^f > 0 \\ -\frac{\left\|\mathcal{W}_{\mathbf{j}}^{\mathbf{f}}\right\|_1}{n}, \text{ if } w_{i,j}^f \leq 0 \end{cases}, w_{i,j}^f \in \mathcal{W}_{\mathbf{j}}^{\mathbf{f}}, w_{i,j}^b \in \mathcal{W}_{\mathbf{j}}^{\mathbf{b}}, i \in [0, C),$$

(3)

where $\mathcal{X}_{\mathbf{j}}^{\mathbf{f}} \in \mathbb{R}^{C \times H \times W}$ and $\mathcal{W}_{\mathbf{j}}^{\mathbf{f}} \in \mathbb{R}^{C_{out} \times C_{in} \times K_h \times K_w}$ are full-precision activations and convolution weights in $j$-th layer, respectively. Similarly, $\mathcal{X}_{\mathbf{j}}^{\mathbf{b}} \in \mathbb{R}^{C \times H \times W}$ and $\mathcal{W}_{\mathbf{j}}^{\mathbf{b}} \in \mathbb{R}^{C_{out} \times C_{in} \times K_h \times K_w}$ are binarized activations and convolution weights in $j$-th layer separately. $x_{i,j}^f$, $w_{i,j}^f$, $x_{i,j}^b$, and $w_{i,j}^b$ are the elements of $i$-th channel of $\mathcal{X}_{\mathbf{j}}^{\mathbf{f}}$, $\mathcal{W}_{\mathbf{j}}^{\mathbf{f}}$, $\mathcal{X}_{\mathbf{j}}^{\mathbf{b}}$, and $\mathcal{W}_{\mathbf{j}}^{\mathbf{b}}$ respectively. $\alpha_{i,j}$ is the learnable coefficient controlling the threshold of sign function for $i$-th channel of $\mathcal{X}_{\mathbf{j}}^{\mathbf{f}}$. It is notable that the weight binarization method is inherited from XONR-Net (Rastegari et al., 2016), of which $\frac{\left\|\mathcal{W}_j^f\right\|_1}{n}$ is the average of absolute weight values and acts as a scaling factor to minimize the difference between binary and full-precision convolution weights.

Then, we use the RPReLU (Liu et al., 2020) as our activation function, which is defined as follows:

$$f(y_{i,j}) = \begin{cases} y_{i,j} - \gamma_{i,j} + \zeta_{i,j}, \text{ if } y_{i,j} > \gamma_{i,j} \\ \beta_{i,j}(y_{i,j} - \gamma_{i,j}) + \zeta_{i,j}, \text{ if } y_{i,j} \leq \gamma_{i,j} \end{cases}, y_{i,j} \in \mathcal{Y}_j, i \in [0, C), \quad (4)$$

where $\mathcal{Y}_{\mathbf{j}} \in \mathbb{R}^{C \times H \times W}$ is the input feature maps of RPReLU function $f(.)$ in $j$-th layer. $y_{i,j}$ is the element of $i$-th channel of $\mathcal{Y}_{\mathbf{j}}$. $\gamma_{i,j}$ and $\zeta_{i,j}$ are learnable shifts for moving the distribution. $\beta_{i,j}$ is a learnable coefficient controlling the slope of the negative part, which acts on $i$-th channel of $\mathcal{Y}_{\mathbf{j}}$.

Different from the common residual block, which consists of two convolutions used in full-precision IR network (Lim et al., 2017), we find that residual connection is essential for binary convolution to supplement the information loss caused by binarization. Thus, we set a residual connection for each binary convolution. Therefore, the BBCU-V1 can be expressed mathematically as:

$$\mathcal{X}_{\mathbf{j+1}}^{\mathbf{f}} = f\left(\mathrm{BatchNorm}\left(\mathcal{X}_{\mathbf{j}}^{\mathbf{b}} \otimes \mathcal{W}_{\mathbf{j}}^{\mathbf{b}}\right) + \mathcal{X}_{\mathbf{j}}^{\mathbf{f}}\right) = f\left(\kappa_{\mathbf{j}}\left(\mathcal{X}_{\mathbf{j}}^{\mathbf{b}} \otimes \mathcal{W}_{\mathbf{j}}^{\mathbf{b}}\right) + \tau_{\mathbf{j}} + \mathcal{X}_{\mathbf{j}}^{\mathbf{f}}\right), \quad (5)$$

where $\kappa_{\mathbf{j}}, \tau_{\mathbf{j}} \in \mathbb{R}^C$ are learnable parameters of BatchNorm in $j$-th layer.

In BNNs, the derivative of the sign function in Eq. 2 is an impulse function that cannot be utilized in training. Thus, we adopt the approximated derivative function as the derivative of the sign function. It can be expressed mathematically as:

$$\text{Approx} \left( \frac{\partial \, \text{Sign} \left( x_i^f \right)}{\partial x_i^f} \right) = \begin{cases} 2 + 2 \left( x_i^f - \alpha_i \right) & \text{if } \alpha_i - 1 \leqslant x_i^f < \alpha_i \\ 2 - 2 \left( x_i^f - \alpha_i \right) & \text{if } \alpha_i \leqslant x_i^f < \alpha_i + 1 \\ 0 & \text{otherwise} \end{cases} . \tag{6}$$

However, EDSR (Lim et al., 2017) demonstrated that BN changes the distribution of images, which is harmful to accurate pixel prediction in SR. So, can we also directly remove the BN in BBCU-V1 to obtain BBCU-V2 (Fig. 2 (b))? In BBCU-V1 (Fig. 2 (a)), we can see that the bit counting operation of binary convolution tends to output large value ranges (from -15 to 15). In contrast, residual connection preserves the full-precision information, which flows from the front end of IR network with a small value range from around -1 to 1. By adding a BN, its learnable parameters can narrow the value range of binary convolution and make it close to the value range of residual connection to avoid full-precision information being covered. The process can be expressed as:

$$\text{Mean} \left( \left| \kappa_{\mathbf{j}} \left( \mathcal{X}_{\mathbf{j}}^{\mathbf{b}} \otimes \mathcal{W}_{\mathbf{j}}^{\mathbf{b}} \right) + \tau_{\mathbf{j}} \right| \right) \rightarrow \text{Mean} \left( \left| \mathcal{X}_{\mathbf{j}}^{\mathbf{f}} \right| \right), \tag{7}$$

where $\kappa_{\mathbf{j}}, \tau_{\mathbf{j}} \in \mathbb{R}^C$ are learnable parameters of BN in $j$-th layer. Thus, compared with BBCU-V1, BBCU-V2 simply removes BN and suffers a huge performance drop.

After the above exploration, we know that BN is essential for BBCU, because it can balance the value range of binary convolution and residual connection. However, BN changes image distributions limiting restoration. In BBCU-V3, we propose residual alignment (RA) scheme by multiplying the value range of input image by an amplification factor $k$ ($k > 1$) to remove BN Figs. 2(c) and 3(b):

$$\text{Mean} \left( \left| \mathcal{X}_{\mathbf{j}}^{\mathbf{b}} \right| \right) \leftarrow \text{Mean} \left( \left| k \mathcal{X}_{\mathbf{j}}^{\mathbf{f}} \right| \right). \tag{8}$$

We can see from the Eq. 8, since the residual connection flows from the amplified input image, the value range of $\mathcal{X}_{\mathbf{j}}^{\mathbf{f}}$ also is amplified, which we define as $k\mathcal{X}_{\mathbf{j}}^{\mathbf{f}}$. Meanwhile, the values of binary convolution are almost not affected by RA, because $\mathcal{X}_{\mathbf{j}}^{\mathbf{b}}$ filters amplitude information of $k\mathcal{X}_{\mathbf{j}}^{\mathbf{f}}$. Different from BatchNorm, RA makes the value range of residual connection close to binary convolution (-60 to 60). Besides, we find that using RA to remove the BatchNorm has two main benefits: **(1)** Similar to full-precision IR networks, the binarized IR networks would have better performance without BatchNorm. **(2)** The structure of BBCU becomes more simple, efficient, and strong.

Based on the above findings, we are aware that the activation function (Eq. 4) in BBCU-V3 (Fig. 2(c)) narrows the negative value range of residual connection, which means it loses negative full-precision information. Thus, we further develop BBCU-V4 by moving the activation function into the residual connection to avoid losing the negative full-precision information. The experiments (Sec. 4.5) show that our design is accurate. We then take BBCU-V4 as the final design of BBCU.

## 3.2 Arm IR Networks with BBCU

As shown in Fig. 3(a), the existing image restoration (IR) networks could be divided into four parts: head $\mathcal{H}$, body $\mathcal{B}$, upsampling $\mathcal{U}$, and tail $\mathcal{T}$. If the IR networks do not need increasing resolution (Zhang et al., 2017), it can remove the upsampling $\mathcal{U}$ part. Specifically, given an LQ input $I_{LQ}$, the process of IR network restoring HQ output $\hat{I}_{HQ}$ can be formulated as:

$$\hat{I}_{HQ} = \mathcal{T}(\mathcal{U}(\mathcal{B}(\mathcal{H}(I_{LQ})))). \tag{9}$$

Previous BNN SR works (Xin et al., 2020; Jiang et al., 2021; Zhang et al., 2021b) concentrate on binarizing the body $\mathcal{B}$ part. However, upsampling $\mathcal{U}$ part accounts for 52.3% total calculations and is essential to be binarized. Besides, the binarized head $\mathcal{H}$ and tail $\mathcal{T}$ parts are also worth exploring.

As shown in Fig. 3(b), we further design different variants of BBCU for these four parts. **(1)** For the head $\mathcal{H}$ part, its input is $I_{LQ} \in \mathbb{R}^{3 \times H \times W}$ and the binary convolution output with $C$ channels. Thus, we cannot directly add $I_{LQ}$ to the binary convolution output for the difference in the number of channels. To address the issue, we develop BBCU-H by repeating $I_{LQ}$ to have $C$ channels. **(2)** For body $\mathcal{B}$ part, since the input and output channels are the same, we develop BBCU-B by directly adding the input activation to the binary convolution output. **(3)** For upsampling $\mathcal{U}$ part, we develop BBCU-U by repeating the channels of input activations to add with the binary convolution. **(4)** For the tail $\mathcal{T}$ part, we develop BBCU-T by adopting $I_{LQ}$ as the residual connection. To better evaluate

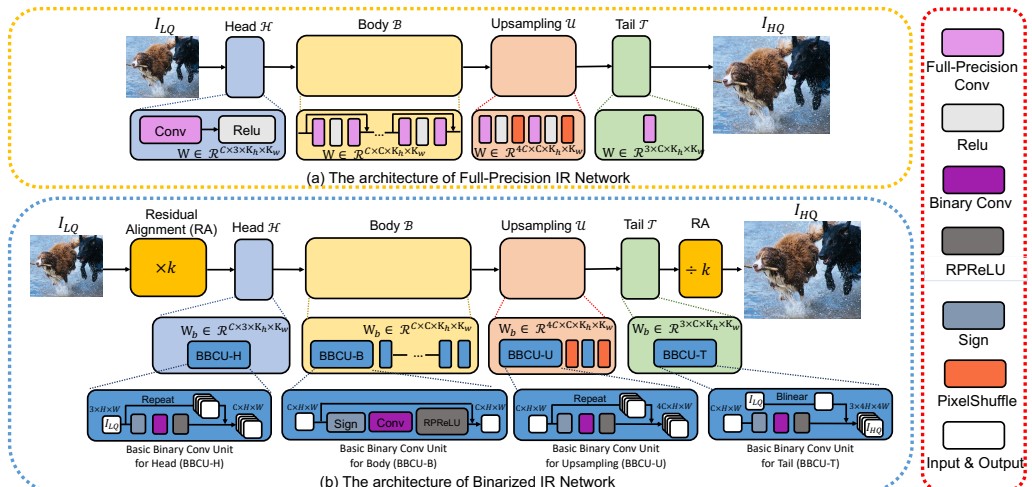

Figure 3: The illustration of full-precision and binary IR networks. (a) The IR network can be generally divided into four parts: head, body, upsampling, and tail. Notably, for IR tasks whose resolution remains unchanged, such as denoising and deblocking, we can ignore the upsampling part. (b) We equip different parts of binarized IR networks with the variants of BBCU.

the benefits of binarizing each part on computations and parameters, we define two metrics:

$$V_C = \left(\text{PSNR}^f - \text{PSNR}^b\right) / \left(\text{OPs}^f - \text{OPs}^b\right), \tag{10}$$

$$V_P = \left(\text{PSNR}^f - \text{PSNR}^b\right) / \left(\text{Parms}^f - \text{Parms}^b\right), \tag{11}$$

where $\text{PSNR}^b$, $\text{OPs}^b$, and $\text{Parms}^b$ denote the performance, calculations, and parameters after binarizing one part of networks. Similarly, $\text{PSNR}^f$, $\text{OPs}^f$, and $\text{Parms}^f$ measure full-precision networks.

We adopt reconstruction loss $L_{rec}$ to guide the image restoration training, which is defined as:

$$L_{rec} = \left\| I_{HQ} - \hat{I}_{HQ} \right\|_1, \tag{12}$$

where $I_{HQ}$ and $\hat{I}_{HQ}$ are the real and restored HQ images, respectively. $\| \cdot \|_1$ denotes the $L_1$ norm.

## 4 EXPERIMENTS

### 4.1 EXPERIMENT SETTINGS

**Training Strategy.** We apply our method to three typical image restoration tasks: image super-resolution, color image denoising, and image compression artifacts reduction. We train all models on DIV2K (Agustsson & Timofte, 2017), which contains 800 high-quality images. Besides, we adopt widely used test sets for evaluation and report PSNR and SSIM. For image super-resolution, we take simple and practical SRResNet (Ledig et al., 2017) as the backbone. The mini-batch contains 16 images with the size of 192×192 randomly cropped from training data. We set the initial learning rate to $1\times10^{-4}$, train models with 300 epochs, and perform halving every 200 epochs. For image denoising and compression artifacts reduction, we take DnCNN and DnCNN3 as backbone (Zhang et al., 2017). The mini-batch contains 32 images with the size of 64×64 randomly cropped from training data. We set the initial learning rate to $1\times10^{-4}$, train models with 300,000 iterations, and perform halving per 200,000 iterations. The amplification factor $k$ in the residual alignment is set to 130. We implement our models with a Tesla V100 GPU.

**OPs and Parameters Calculation of BNN.** Following Rastegari et al. (2016); Liu et al. (2018), the operations of BNN ($\text{OPs}^b$) is calculated by $\text{OPs}^b = \text{OPs}^f / 64$ ($\text{OPs}^f$ indicates FLOPs), and the parameters of BNN ($\text{Parms}^b$) is calculated by $\text{Parms}^b = \text{Parms}^f / 32$.

### 4.2 EVALUATION ON IMAGE SUPER-RESOLUTION

Following BAM (Xin et al., 2020), we take SRResNet (Ledig et al., 2017) as backbone. We binarize body $\mathcal{B}$ part of SRResNet with some SOTA BNNs including BNN (Courbariaux et al., 2016), Bi-Real (Liu et al., 2018), IRNet (Qin et al., 2020), ReActNet (Liu et al., 2020), and BAM (Xin et al., 2020), BTM (Jiang et al., 2021). For comparison, we adopt our BBCU to binarize body $\mathcal{B}$ part to obtain BBCU-L. Furthermore, we binarize body $\mathcal{B}$ and upsampling $\mathcal{U}$ parts simultaneously to develop

Table 1: Quantitative comparison (average PSNR/SSIM) with BNNs for classical **image Super-Resolution** on benchmark datasets. Best and second best performance among BNNs are in red and blue colors, respectively. OPs are computed based on LQ images with a resolution of 320×180.

| Methods | Scale | Ops (G) | Params (K) | Set5 PSNR | Set5 SSIM | Set14 PSNR | Set14 SSIM | B100 PSNR | B100 SSIM | Urban100 PSNR | Urban100 SSIM | Manga109 PSNR | Manga109 SSIM |
|---|---|---|---|---|---|---|---|---|---|---|---|---|---|
| SRResNet | | 85.43 | 1367 | 38.00 | 0.9605 | 33.59 | 0.9171 | 32.19 | 0.8997 | 32.11 | 0.9282 | 38.56 | 0.9770 |
| SRResNet-Lite | | 3.08 | 49 | 37.21 | 0.9578 | 32.86 | 0.9113 | 31.67 | 0.8936 | 30.48 | 0.9109 | 36.70 | 0.9722 |
| Bicubic | | - | - | 33.97 | 0.9330 | 30.55 | 0.8750 | 29.73 | 0.8494 | 27.07 | 0.8456 | 31.24 | 0.9383 |
| BNN | | 18.55 | 225 | 32.25 | 0.9118 | 29.25 | 0.8406 | 28.68 | 0.8104 | 25.96 | 0.8088 | 29.16 | 0.9127 |
| Bi-Real | | 18.55 | 225 | 32.32 | 0.9123 | 29.47 | 0.8424 | 28.74 | 0.8111 | 26.35 | 0.8161 | 29.64 | 0.9167 |
| BAM | ×2 | 20.52 | 226 | 37.21 | 0.9560 | 32.74 | 0.9100 | 31.6 | 0.8910 | 30.20 | 0.9060 | - | - |
| IRNet | | 18.55 | 225 | 37.27 | 0.9579 | 32.92 | 0.9115 | 31.76 | 0.8941 | 30.63 | 0.9122 | 36.77 | 0.9724 |
| BTM | | 18.55 | 225 | 37.22 | 0.9575 | 32.93 | 0.9118 | 31.77 | 0.8945 | 30.79 | 0.9146 | 36.76 | 0.9724 |
| ReActNet | | 18.55 | 225 | 37.26 | 0.9579 | 32.97 | 0.9124 | 31.81 | 0.8954 | 30.85 | 0.9156 | 36.92 | 0.9728 |
| BBCU-M (Ours) | | 1.83 | 46 | 37.44 | 0.9584 | 33.04 | 0.9127 | 31.81 | 0.8946 | 30.84 | 0.9149 | 37.20 | 0.9738 |
| BBCU-L (Ours) | | 18.55 | 225 | 37.58 | 0.9590 | 33.18 | 0.9143 | 31.91 | 0.8962 | 31.12 | 0.9179 | 37.50 | 0.9746 |
| SRResNet | | 146.14 | 1515 | 32.16 | 0.8951 | 28.60 | 0.7822 | 27.58 | 0.7364 | 26.11 | 0.7870 | 30.46 | 0.9089 |
| SRResNet-Lite | | 5.39 | 54 | 31.40 | 0.8843 | 28.11 | 0.7692 | 27.26 | 0.7243 | 25.19 | 0.7544 | 28.92 | 0.8863 |
| Bicubic | | - | - | 28.63 | 0.8128 | 26.21 | 0.7087 | 26.04 | 0.6719 | 23.24 | 0.6114 | 25.07 | 0.7904 |
| BNN | | 79.20 | 372 | 27.56 | 0.7896 | 25.51 | 0.6820 | 25.54 | 0.6466 | 22.68 | 0.6352 | 24.19 | 0.7670 |
| Bi-Real | | 79.20 | 372 | 27.75 | 0.7935 | 25.79 | 0.6879 | 25.59 | 0.6478 | 22.91 | 0.6450 | 24.57 | 0.7752 |
| BAM | ×4 | 81.17 | 373 | 31.24 | 0.8780 | 27.97 | 0.7650 | 27.15 | 0.7190 | 24.95 | 0.7450 | - | - |
| IRNet | | 79.20 | 372 | 31.38 | 0.8835 | 28.08 | 0.7679 | 27.24 | 0.7227 | 25.21 | 0.7536 | 28.97 | 0.8863 |
| BTM | | 79.20 | 372 | 31.43 | 0.8850 | 28.16 | 0.7706 | 27.29 | 0.7256 | 25.34 | 0.7605 | 29.19 | 0.8912 |
| ReActNet | | 79.20 | 372 | 31.54 | 0.8859 | 28.19 | 0.7705 | 27.31 | 0.7252 | 25.35 | 0.7603 | 29.25 | 0.8912 |
| BBCU-M (Ours) | | 3.95 | 51 | 31.54 | 0.8862 | 28.20 | 0.7718 | 27.31 | 0.7263 | 25.35 | 0.7602 | 29.22 | 0.8918 |
| BBCU-L (Ours) | | 79.20 | 372 | 31.79 | 0.8905 | 28.38 | 0.7762 | 27.41 | 0.7303 | 25.62 | 0.7696 | 29.69 | 0.8992 |

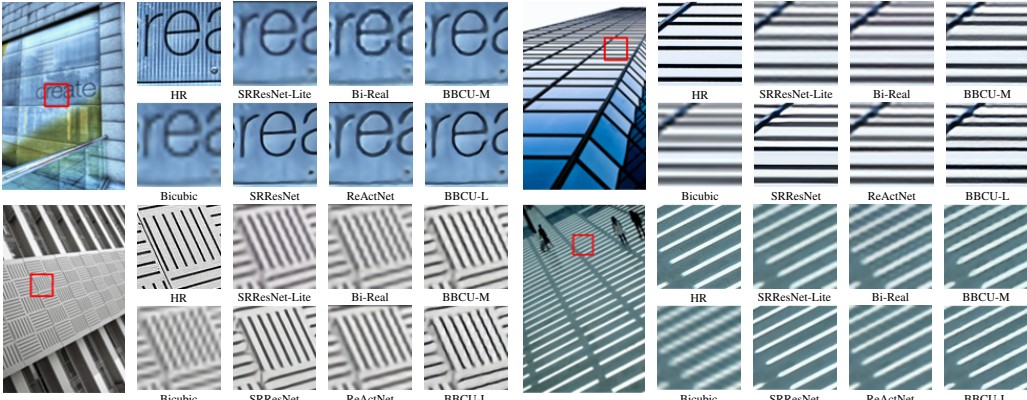

Figure 4: Visual comparison of BNNs for 4× image super-resolution.

BBCU-M. Besides, we reduce the number of channels of SRResNet to 12, obtaining SRResNet-Lite. In addition, we use Set5 (Bevilacqua et al., 2012), Set14 (Zeyde et al., 2010), B100 (Martin et al., 2001), Urban100 (Huang et al., 2015), and Manga109 (Matsui et al., 2017) for evaluation.

The quantitative results (PSNR and SSIM), the number of parameters, and operations of different methods are shown in Tab. 1. Compared with other binarized methods, our BBCU-L achieves the best results on all benchmarks and all scale factors. Specifically, for 4× SR, our BBCU-L surpasses ReActNet by 0.25dB, 0.27dB, and 0.44dB on Set5, Urban100, and Manga109, respectively. For 2× SR, BBCU-L also achieves 0.32dB, 0.27dB, and 0.58dB improvement on these three benchmarks compared with ReActNet. Furthermore, our BBCU-M achieves the second best performance on most benchmarks consuming 5% of operations and 13.7% parameters of ReActNet (Liu et al., 2020) on 4× SR. Besides, BBCU-L significantly outperforms SRResNet-Lite by 0.16dB and 0.3dB with less computational cost, showing the superiority of BNN. The qualitative results are shown in Fig. 4, and our BBCU-L has the best visual quality containing more realistic details close to respective ground-truth HQ images. More qualitative results are provided in appendix.

### 4.3 EVALUATION ON IMAGE DENOISING

For image denoising, we use DnCNN (Zhang et al., 2017) as the backbone and binarize its body $\mathcal{B}$ part with BNN methods, including BNN, Bi-Real, IRNet, BTM, ReActNet, and our BBCU. We also develop DnCNN-Lite by reducing the number of channels of DnCNN from 64 to 12. The standard benchmarks: Urban100 (Huang et al., 2015), BSD68 (Martin et al., 2001), and Set12 (Shan et al.,

Table 2: Quantitative comparison (average PSNR) with BNNs for classical **image denoising** on benchmark datasets. Best and second best performance among BNNs are in red and blue colors, respectively. OPs is computed based on LQ images with a resolution of 320×180.

| Methods | OPs (G) | Params (K) | CBSD68 | | | Kodak24 | | | Urban100 | | |
|---|---|---|---|---|---|---|---|---|---|---|---|
| | | | $\sigma = 15$ | $\sigma = 25$ | $\sigma = 50$ | $\sigma = 15$ | $\sigma = 25$ | $\sigma = 50$ | $\sigma = 15$ | $\sigma = 25$ | $\sigma = 50$ |
| DnCNN | 38.42 | 667 | 33.90 | 31.24 | 27.95 | 34.60 | 32.14 | 28.95 | 32.98 | 30.81 | 27.59 |
| DnCNN-Lite | 1.38 | 24 | 32.26 | 29.64 | 26.49 | 32.73 | 30.25 | 27.22 | 30.97 | 28.46 | 25.21 |
| BNN | 0.8 | 24 | 26.90 | 22.67 | 16.41 | 27.12 | 22.58 | 16.25 | 26.67 | 22.67 | 16.54 |
| Bi-Real | 0.8 | 24 | 30.73 | 28.72 | 25.63 | 30.97 | 29.17 | 26.11 | 30.18 | 28.18 | 25.11 |
| IRNet | 0.8 | 24 | 31.37 | 29.01 | 26.75 | 31.61 | 29.54 | 27.48 | 30.57 | 28.35 | 26.00 |
| BTM | 0.8 | 24 | 32.25 | 29.91 | 26.79 | 32.75 | 30.64 | 27.62 | 30.99 | 29.05 | 25.86 |
| ReActNet | 0.8 | 24 | 32.26 | 29.95 | 26.93 | 32.78 | 30.65 | 27.70 | 31.27 | 29.20 | 26.01 |
| BBCU (Ours) | 0.8 | 24 | 33.08 | 30.56 | 27.33 | 33.66 | 31.28 | 28.15 | 32.27 | 29.96 | 26.61 |

Table 3: Quantitative comparison (average PSNR/SSIM) with BNNs for classical **JPEG compression artifact reduction**. Best and second best performance among BNNs are in red and blue colors, respectively. OPs is computed based on LQ images with a resolution of 320×180.

| Methods | OPs (G) | Params (K) | Live1 | | | | Classic5 | | | |
|---|---|---|---|---|---|---|---|---|---|---|
| | | | $q = 10$ | $q = 20$ | $q = 30$ | $q = 40$ | $q = 10$ | $q = 20$ | $q = 30$ | $q = 40$ |
| DnCNN-3 | 38.29 | 665 | 29.19/0.8123 | 31.59/0.8802 | 32.98/0.9090 | 33.96/0.9247 | 29.40/0.8026 | 31.63/0.8610 | 32.91/0.8861 | 33.77/0.9003 |
| Dncnn-3-Lite | 1.36 | 24 | 28.90/0.8061 | 31.27/0.8746 | 32.62/0.9029 | 33.52/0.9179 | 29.80/0.8000 | 32.09/0.8611 | 33.39/0.8868 | 34.25/0.9011 |
| BNN | 0.66 | 22 | 28.48/0.7925 | 30.82/0.8661 | 32.19/0.8965 | 33.12/0.9128 | 29.41/0.7879 | 31.71/0.8537 | 33.04/0.8817 | 33.89/0.8964 |
| Bi-Real | 0.66 | 22 | 28.67/0.8011 | 31.06/0.8706 | 32.42/0.8998 | 33.36/0.9158 | 29.54/0.7934 | 31.88/0.8573 | 33.21/0.8845 | 34.08/0.8996 |
| IRNet | 0.66 | 22 | 28.73/0.8019 | 31.13/0.8704 | 32.49/0.8993 | 33.40/0.9148 | 29.63/0.7947 | 31.96/0.8570 | 33.28/0.8836 | 34.12/0.8983 |
| BTM | 0.66 | 22 | 28.75/0.8032 | 31.18/0.8725 | 32.51/0.9005 | 33.38/0.9153 | 29.65/0.7968 | 32.02/0.8597 | 33.30/0.8853 | 34.11/0.8991 |
| ReActNet | 0.66 | 22 | 28.81/0.8025 | 31.20/0.8709 | 32.52/0.8981 | 33.37/0.9123 | 29.70/0.7956 | 32.04/0.8581 | 33.32/0.8831 | 34.11/0.8964 |
| BBCU (Ours) | 0.66 | 22 | 29.06/0.8087 | 31.43/0.8780 | 32.80/0.9067 | 33.75/0.9221 | 30.00/0.8028 | 32.27/0.8645 | 33.59/0.8903 | 34.45/0.9044 |

2019) are applied to evaluate each method. Additive white Gaussian noises (AWGN) with different noise levels $\sigma$ (15, 25, 50) are added to the clean images.

The quantitative results of image denoising are shown in Tab. 2, respectively. As one can see, our BBCU achieves the best performance among compared BNNs. In particular, our BBCU surpasses the state-of-the-art BNN model ReActNet by 0.82dB, 0.88dB, and 1dB on CBSD68, Kodak24, and Urban100 datasets respectively as $\sigma = 15$. Compared with DnCNN-Lite, our BBCU surpasses it by 0.92dB, 1.03dB, and 1.5dB on these three benchmarks as $\sigma = 15$ consuming 58% computations of DnCNN-Lite. Qualitative results are provided in appendix.

## 4.4 EVALUATION ON JPEG COMPRESSION ARTIFACT REDUCTION

For this JPEG compression deblocking, we use practical DnCNN-3 (Zhang et al., 2017) as the backbone and replace the full-precision body $\mathcal{B}$ part of DnCNN3 with some competitive BNN methods, including BNN, Bi-Real, IRNet, BTM, ReActNet, and our BBCU. The compressed images are generated by Matlab standard JPEG encoder with quality factors $q \in \{10, 20, 30, 40\}$. We take the widely used LIVE1 (Sheikh, 2005) and Classic5 (Foi et al., 2007) as test sets to evaluate the performance of each method. The quantitative results are presented in Tab. 3. As we can see, our BBCU achieves the best performance on all test sets and quality factors among all compared BNNs. Specifically, our BBCU surpasses the state-of-the-art BNN model ReActNet by 0.38dB and 0.34dB on the Live1 and Classic5 as $q = 40$. In addition, our BBCU surpasses DnCNN-3-Lite by 0.16dB and 0.2dB on benchmarks as $q = 10$. The visual comparisons are provided in appendix.

## 4.5 ABLATION STUDY

**Basic Binary Convolution Unit.** To validate BBCU for the binarized IR network, we binarize the body part of SRResNet with four variants of BBCU (Fig. 2) separately. The results are shown in Tab. 4. **(1)** Compared with BBCU-V1, BBCU-V2 declines by 0.14dB and 0.24dB

Table 4: PSNR (dB) values (4×) on four types of basic binary convolution unit (BBCU).

| Methods | OPs (G) | Set5 | Set14 | B100 | Urban100 | Manga109 |
|---|---|---|---|---|---|---|
| BBCU-V1 | 79.20 | 31.54 | 28.19 | 27.31 | 25.35 | 29.25 |
| BBCU-V2 | 79.20 | 31.37 | 28.07 | 27.22 | 25.21 | 29.01 |
| BBCU-V3 | 79.20 | 31.71 | 28.31 | 27.37 | 25.51 | 29.54 |
| BBCU-V4 | 79.20 | 31.79 | 28.38 | 27.41 | 25.62 | 29.69 |

on Urban100 and Manga109. This is because BBCU-V2 simply removes the BN making the value range of binary Conv far larger than residual connection and cover full-precision information (Fig. 2). **(2)** BBCU-V3 adds residual alignment scheme on BBCU-V2, which addresses the value range imbalance between binary Conv and residual connection and removes the BN. Since the BN is harmful for IR networks, BBCU-V3 surpasses BBCU-V1 by 0.17dB, 0.16dB, and 0.29dB on Set5, Urban100, and Manga109 respectively. **(3)** BBCU-V4 moves the activation function into the residual connection, which preserves the full-precision negative values of residual connection (Fig. 2). Thus, BBCU-V4 outperforms BBCU-V3.

Table 5: The breakpoint position of residual connection.

| Position Idx | PSNR$_b$ (dB) |
|---|---|
| 1 | 25.43 |
| 5 | 25.44 |
| 10 | 25.48 |
| 15 | 25.46 |
| 20 | 25.45 |
| 25 | 25.44 |
| - | 25.62 |

Table 6: The binarization benefit of different parts in IR networks. We test models on Set14, and PSNR$^f$ is 28.60dB.

| | Head $\mathcal{H}$ | Body $\mathcal{B}$ | Upsampling $\mathcal{U}$ | Tail $\mathcal{T}$ |
|---|---|---|---|---|
| OPs$^f$ (M) | 99.53 | 67947.73 | 76441.19 | 1592.53 |
| OPs$^b$ (M) | 1.56 | 1061.68 | 1194.39 | 24.88 |
| Params$_f$ (K) | 1.73 | 1179.65 | 294.91 | 1.73 |
| Params$_b$ (K) | 0.05 | 36.86 | 9.22 | 0.05 |
| PSNR$^b$ (dB) | 28.58 | 28.38 | 28.59 | 27.76 |
| $V_C \downarrow (\times 10^{-6})$ | 204.14 | 3.29 | 0.13 | 535.83 |
| $V_P \downarrow (\times 10^{-3})$ | 11.91 | 0.19 | 0.04 | 500.00 |

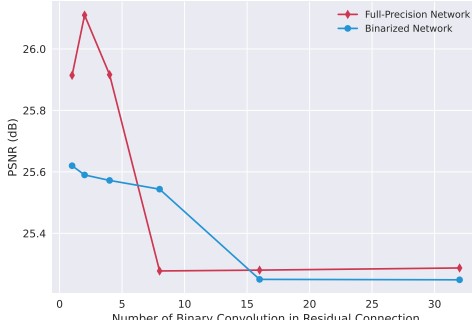

Figure 5: The effect of number of binary convolution in residual connection on SRResNet.

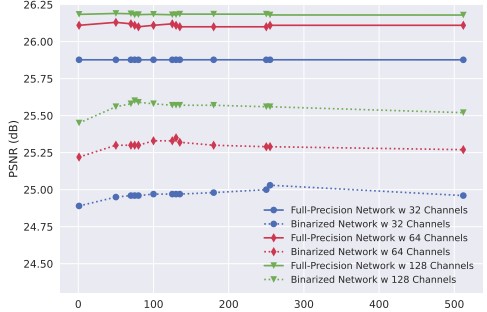

Figure 6: The effect of amplification factor on SRResNet with various number of channels.

**Residual Connection.** We take SRResNet as the backbone with 32 Convs in body part and explore the relationship between performance and the number of binary convolution in residual connection. Specifically, for both full-precision and binarized SRResNet, we set 1,2,4,8,16, and 32 Convs with a residual connection as basic convolution units, respectively. We evaluate SRResNet equipped with these basic convolution units on $4\times$ Urban100 (see Fig. 5). **(1)** For full-precision networks, it is best to have 2 Convs with a residual connection to form a basic convolution unit. Besides, if there are more than 4 Convs in a basic convolution unit, its performance would sharply drop. For binarized networks, it is best to equip each binary convolution with a residual connection. In addition, compared with the full-precision network, the binarized network is more insensitive to the growth of the Convs number in the basic convolution unit. **(2)** For binarized SRResNet, we delete residual connection of a BBCU in a certain position. In Tab. 5, it seems that once the residual connection is removed (the full-precision information is lost) at any position, the binarized SRResNet would suffer a severe performance drop.

**Amplification Factor.** As shown in Fig. 6, the performance of full-precision remain basically unchanged with the variation of amplification factor $k$. However, the binarized network is sensitive to the $k$. The best $k^*$ is related to the number of channels $n$, which empirically fits $k^* = 130n/64$. Intuitively, the larger number of channels makes binary convolution count more elements and have larger results, which needs larger $k$ to increase the value of residual connection for balance.

**The Binarization Benefit for Different Parts of IR Network.** We binarize one part in SRResNet with BBCU in Fig. 3 while keeping other parts full-precision. The results are shown in Tab. 6. We use $V_C$ (Eq. 10) and $V_P$ (Eq. 11) as metric to value binarization benefit of different parts. We can see that Upsampling part is most worth to be binarized. However, it is ignored by previous works (Xin et al., 2020). The binarization benefit of first and last convolution is relatively low.

## 5 CONCLUSION

This work devotes attention to exploring the performance of BNN on low-level tasks and search of generic and efficient basic binary convolution unit. Through decomposing and analyzing existing elements, we propose BBCU, a simple yet effective basic binary convolution unit, that outperforms existing state-of-the-arts with high efficiency. Furthermore, we divide IR networks to four parts, and specially develop the variants of BBCU for them to explore the binarization benefits. Overall, BBCU provide insightful analyses on BNN and can serve as strong basic binary convolution unit for future binarized IR networks, which is meaningful to academic research and industry.

## ACKNOWLEDGMENTS

This work was partly supported by the Alexander von Humboldt Foundation, the National Natural Science Foundation of China(No. 62171251), the Natural Science Foundation of Guangdong Province(No.2020A1515010711), the Special Foundations for the Development of Strategic Emerging Industries of Shenzhen(Nos.JCYJ20200109143010272 and CJGJZD20210408092804011) and Oversea Cooperation Foundation of Tsinghua.

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
