# OpenReview forum: "Basic Binary Convolution Unit for Binarized Image Restoration Network"
_ICLR.cc/2023/Conference — ICLR 2023 poster_

### Official Review · Reviewer_zzdZ · 2022-10-24

**Confidence:** 4
**Clarity, Quality, Novelty And Reproducibility:** No further concern in this category.
**Correctness:** 3
**Technical Novelty And Significance:** 2
**Empirical Novelty And Significance:** Not applicable
**Recommendation:** 6

**Strength And Weaknesses:**

**Strength**

- The paper is easy to follow.
- Experimental results seem to be strong. The proposed method outperforms all reference methods across all three tasks.

**Weaknesses**

- Limited novelty: The presented findings regarding residual connection, BN and the position of RPrelu are not supervising. Similar observations have already been reported and discussed in Bireal, Real2binaryNet and ReActNet papers. In terms of network design, there are only minor adaptions on top of ReActNet block design making the proposed BBCU block incremental. The current design lacks significant innovation.

- Current design choices are more based on empirical attempts and lack of refinement of optimization methods specified on IR tasks. For example, BNN-specific training methods such as two-stage training (real2binaryNet), progressive sign function ([1]), knowledge distillation training, etc., have not been verified in this work. However, these methods show significant improvement in the image classification task.

- If we want to demonstrate the practical effectiveness of BNN-based IR methods, it will be better to also compare with more compression methods, such as pruning-based methods and compact model designs. Also, is it possible to use a binary inference engine like larq or Dabnn to verify the real inference speed on Arm cpu? We know that the test results on hardware can sometimes be quite different from the theoretical computational complexity OPs. Arms cpu is also an important computing platform for IR tasks.

[1] Gradient matters: Designing binarized neural networks via enhanced information-flow

**Summary Of The Paper:**

This paper evaluated some essential design choices of existing binary neural networks for image restoration tasks. The authors further propose a network design and evaluate it on three different tasks. Experimental results show that the proposed binary network outperforms all the reference methods.

**Summary Of The Review:**

The paper is well structured, and the experimental result seems to be strong. But, I still have several concerns regarding novelty, a more complete comparison, and adding inference speed evaluation.

---

> ### Author Response · Authors · 2022-11-16
> **Author feedback to Reviewer zzdZ**
>
> **Q1:  Similar observations have already been reported and discussed in Bireal, Real2binaryNet and ReActNet papers. In terms of network design, there are only minor adaptions on top of ReActNet block design making the proposed BBCU block incremental.**
>
> A1: Bi-Real, Real to binaryNet, and ReActNet are designed for image classification, while our BBCU is developed for image restoration. Since there are significant differences between these two tasks, our empirical observations and findings are new and important for image restoration.  Directly applying this SOTA classification BNNs achieve inferior performance. For instance, our method achieves significant improvements over ReActNet and Bi-Real (please kindly see Tables 1, 2, and 3).
>
> Differences with ReActNet:
> Our BBCU block differs from ReActNet blocks in several different aspects and our new designs improve the model capability for IR, which can be verified by our experiments. (1) BN is commonly used in previous SoTA BNN-based classification models (e.g., ReActNet). However, BN will decrease IR performance which has been widely demonstrated.  In this paper, we first discover the importance of BN in mitigating value range misalignment between full-precision residual connection and binary Conv, and propose a novel residual alignment to address the misalignment issue with removing BN from IR networks. (2) Taking a common practice in binary residual networks, ReActNet puts the activation function outside the residual connection, However, we find that the activation function will damage the negative full-precision information, which is important for IR. To solve the issue, we design a new BNN  unit, which moves the activation function position into residual connection. (3) These BNN methods including ReActNet cannot be directly applied to all parts of IR networks (e.g., upscaling), which significantly lowers their efficiency. We specially design four variants of BBCU (head, body, upsampling, tail) that can binarize all parts of IR networks.
>
>  In addition, we believe that our key findings are more important and the identified problems are critical for binarized  IR networks. Future research can design better solutions to further address the problems. Our proposed method is only one of the instantiations.
>
> ____
>
> **Q2: BNN-specific training methods such as two-stage training (real2binaryNet), progressive sign function, knowledge distillation training, etc., have not been verified in this work.**
>
> A2: （1）Thanks for your suggestions! We conducted these experiments of these BNN-specific training methods on 4x SR. The results are shown in the following table. We can see that these methods cannot bring a significant improvement to SR tasks. Different from the Semantic understanding of classification, image restoration requires precise pixel prediction. Therefore, compared with enhancing training strategy, our BBCU paying more attention to full-precision information preservation can bring significant gain for IR.
>
> (2) Our design choices are associated with the basic operating rules of BNN, which are well-founded. For example, the residual misalignment is caused by the different Conv forms of residual connection and binary Conv (BC). BC (Eq.1, Fig.7) counts the number of 1 or -1 elements to obtain the results with large values while the value range of full-precision residual connection is close to the input images with the value range of 0 to 1 or -1 to 1.  Moreover, we propose RA to reduce the residual misalignment (Eq.7 and Eq.8 of the paper). And activation function will damage negative full-precision information and we adjust its position to avoid the issue.  Moreover, experiments also show our observations and design choices are useful for various IR tasks.
>
> | Method                             | Ops  | Params | Set5         | Set14        | B100         | Urban100     | Manga109      |   |   |
> |------------------------------------|------|--------|--------------|--------------|--------------|--------------|---------------|---|---|
> | ReActNet                           | 79.2 | 372    | 31.54/0.8859 | 28.19/0.7705 | 27.31/0.7252 | 25.35/0.7603 | 29.25/0.8912  |   |   |
> | ReActNet+ two-stage training       | 79.2 | 372    | 31.56/0.8870 | 28.22/0.7723 | 27.33/0.7272 | 25.43/0.7633 | 29.25/0.8929  |   |   |
> | ReActNet+progressive sign function | 79.2 | 372    | 31.44/0.8851 | 28.15/0.7697 | 27.30/0.7252 | 25.32/0.7588 | 28.99/0.8891  |   |   |
> | ReActNet+KD                        | 79.2 | 372    | 31.30/0.8819 | 28.04/0.7658 | 27.21/0.7205 | 25.16/0.7515 | 28.82/0.8845  |   |   |
> | BBCU-L                             | 79.2 | 372    | 31.79/0.8905 | 28.38/0.7762 | 27.41/0.7303 | 25.62/0.7696 | 29.69/0.8992  |   |   |
> |                                    |      |        |              |              |

---

> ### Author Response · Authors · 2022-11-16
> **Author feedback to Reviewer zzdZ**
>
> **Q3: It will be better to also compare with more compression methods, such as pruning-based methods and compact model designs.**
>
> A3: Thanks for your suggestion! To the best of our knowledge, most existing IR methods are not compressed to the level of binarized IR networks. We apply the structured pruning scheme and knowledge distillation to compress SRResnet to the level of binarized IR networks. The results are shown in the following table. Our BBCU has significant superiority in extreme model compression.
>
> | Method           | OPs  | Params | Set5         | Set14        | B100         | Urban100     | Manga109      |   |   |
> |------------------|------|--------|--------------|--------------|--------------|--------------|---------------|---|---|
> | SRResNet-Lite    | 5.39 | 54     | 31.40/0.8843 | 28.11/0.7692 | 27.26/0.7243 | 25.19/0.7544 | 28.92/0.8863  |   |   |
> | SRResNet-prune   | 5.39 | 54     | 31.41/0.8851 | 28.11/0.7700 | 27.26/0.7255 | 25.22/0.7558 | 28.98/0.8876  |   |   |
> | SRResNet-KD | 5.39 | 54     | 31.45/0.8845 | 28.12/0.7698 | 27.26/0.7249 | 25.23/0.7554 | 28.94/0.8870  |   |   |
> | BBCU-M           | 3.95 | 51     | 31.54/0.8862 | 28.20/0.7718 | 27.31/0.7263 | 25.35/0.7602 | 29.22/0.8918  |   |   |
> |                  |      |        |              |              |              |
>
> ____
>
> **Q4: Verify the real inference speed on Arm CPU.**
>
> A4: Thanks for your suggestion!  We deploy our BBCU-M BBCU-L, ReActNet, SRResNet, and SRResNe-Lite on HUAWEI MAIMANG7 to perform 4x SR with 180x320 LR input. The inference speed of these methods is shown in the following table. We can see that our BBCU can have better inference speed and performance. Since the scarcity of 1-bit hardware, these inference frameworks are unable to achieve the potential of  BNNs on Arm devices. In the future, with the popularity of 1-bit specific inference devices, the BNN can obtain far more significant acceleration.
>
> | method       | Time (ms)  |   |   |   |   |   |   |   |   |
> |--------------|------------|---|---|---|---|---|---|---|---|
> | SRResNet     | 42032.4    |   |   |   |   |   |   |   |   |
> | SRResNe-Lite | 2163.7     |   |   |   |   |   |   |   |   |
> | ReActNet     | 23944.4    |   |   |   |   |   |   |   |   |
> | BBCU-M       | 2142.1     |   |   |   |   |   |   |   |   |
> | BBCU-L       | 23928.5    |   |   |   |   |   |   |   |   |
> |              |            |   |   |   |   |   |   |   |   |

---

> ### Comment · Reviewer_zzdZ · 2022-11-29
> **Thanks for the response and additional results.**
>
> Thanks to the authors for their detailed replies, and further experimental results addressed some of my concerns, I thus increased my rating.

---

> > ### Author Response · Authors · 2022-11-30
> > **Thanks Reviewer zzdZ for approving our work**
> >
> > Dear Reviewer zzdZ,
> >
> > Thanks for your response. We are happy to see that our experimental results can solve your concerns.
> >
> > Best,
> >
> > Authors

---

### Official Review · Reviewer_2H4s · 2022-10-24

**Confidence:** 5
**Correctness:** 3
**Technical Novelty And Significance:** 4
**Empirical Novelty And Significance:** 4
**Recommendation:** 8

**Clarity, Quality, Novelty And Reproducibility:**

This paper is well-organized and easy to follow. IR model compression is an important and urgent research direction and BNN is a promising compression method. It is worth exploring BNN on low-level vision. This article deeply analyzes the components of BNN and proposes a practical basic binary convolution unit for binary IR networks. It provides insightful and interesting analyses and conclusions in experiments to guide later BNN design and the results are impressive and general. Besides, the author provides demo codes to reproduce the results in the paper, which makes the paper more reliable and convincing.

**Strength And Weaknesses:**

Strengths: The paper is easy to follow and provides many insightful analyses, and the results are impressive and show promising potential for BNNs.
1. IR model compression is an important and urgent research direction, and BNN is a promising compression method. This paper tackles this problem and thus potentially can have a big impact.
2. This paper rethinks and analyzes the component of BNN. It is interesting and practical. The visualizations, experiments, and analyses are sound and interesting.
3. The experiment results are impressive and general. The proposed model can significantly improve performance on different low-level vision tasks compared with SOTA methods.
4. The ablation studies are insightful. It shows the binarization benefit of different parts and the performance of CNN and BNN to the residual connection, etc.
5. The paper is well written and clearly organized.

Weakness： I have the following questions about the method and result details.
1. About the method and experiments, do you use the BBCU-V4 in Fig.2 to arm the IR networks? It is better to clarify this point.
2. In Tab.6, are the OPs computed based on 320×180 LQ images?  Besides, I hope the author provides OPs in Tab.4.
3. Typos: In Fig.3 (b), the “BBCM-U“ seems not correct.

**Summary Of The Paper:**

Since the different properties in CNN and BNN, it hardly applies the experience of designing CNN to develop BNN. In this paper, the author rethinks and analyzes the actual function of components in binary convolution, such as residual connection, BatchNorm, activation function, and structure, for IR tasks. Based on their observations and analysis, they designed a general basic binary convolution unit for binarized image restoration networks. Extensive results can support their claimed contributions.

**Summary Of The Review:**

The paper is well written. The paper deeply analyzes the components of BNN and designs a BBCU. The ablation studies, like the visualization of feature distribution, demonstrate the effect of the method. The main comparison results with others are impressive and general. Overall, this paper is interesting, insightful, and practical.

---

> ### Author Response · Authors · 2022-11-16
> **Author feedback to Reviewer 2H4s**
>
> **Q1: Do you use the BBCU-V4 in Fig.2 to arm the IR networks? It is better to clarify this point.**
>
> A1: Yes, we use BBCU-V4 in Fig.2. We have noted this point in Fig.2 in the revised paper.
>
> ___
>
> **Q2: In Tab.6, are the OPs computed based on 320×180 LQ images?**
>
> A2: Yes, in all experiments in the paper,  OPs are computed based on 320×180 LQ images.
>
> ___
>
> **Q3: I hope the author provides OPs in Tab.4.**
>
> A3: Thanks for your suggestion! All the OPs of compared BBCU in Tab.4 are 79.20 G. We have updated the results in Tab.4 of the revised paper.
>
>  ___
>
> **Q4: Typos: In Fig.3 (b), the “BBCM-U“ seems not correct.**
>
> A4: Thanks for your suggestion! We have revised ”BBCM-U” to ”BBCU-U” in Fig 3 (b) of the revised paper.

---

> > ### Comment · Reviewer_2H4s · 2022-11-29
> > **Response**
> >
> > Thanks for your response which solves my concerns. I like your idea, and the paper is clear and seems to be reproduced easily. I think BNN is a very feasible and promising compression method to help the IR models adapt to resource-limited devices, which is worth exploring. Your observations and optimization of BBCU for binary IR are interesting and supported by extensive experiments, visualizations, and analyses. The results are impressive and obtain significant and general improvements in binary IR tasks. I think your observations and BBCU are reasonable and general for binary IR, which can inspire others to investigate more and deeper for binary IR. Many additional results (including demo code) are provided in the supplementary file further making this work solid.
> >
> > I also refer to other reviewers’ comments and the authors’ corresponding responses.  I would like to keep my initial score and vote for acceptance.

---

> > > ### Author Response · Authors · 2022-11-30
> > > **Thanks Reviewer 2H4s for approving our work**
> > >
> > > Dear Reviewer 2H4s,
> > >
> > > Thanks for agreeing with the idea, experiments, clarity, and potential of our work. We are happy to see that our revised paper  can solve your concerns.
> > >
> > > Best,
> > >
> > > Authors

---

### Official Review · Reviewer_WV2K · 2022-10-25

**Confidence:** 5
**Correctness:** 2
**Technical Novelty And Significance:** 2
**Empirical Novelty And Significance:** 2
**Recommendation:** 3

**Clarity, Quality, Novelty And Reproducibility:**

The paper lacks clarity and needs a lot of work to improve. Please note the weaknesses section in the review for further points on this. Novelty can't be determined with certainty as reproducibility of the work is lacking.

**Details Of Ethics Concerns:**

No ethical concerns as of yet.

**Strength And Weaknesses:**

Strengths::
1) The motivation case study in section 1 is good.
2) The experimental section is good but can be improved based on the point mentioned in the weaknesses section.

Weaknesses::
The paper can be improved based on the following points.
1) The section 1 can be rearranged and written better. For example, analysis are provided in numerate bullet points but the paragraphing for each analysis is not consistent. Please rectify that for better readability.
2) It would be better to include some form of content table for better readability at the end of section 1 to describe the progression of the paper texts. Readability of the paper is not good and needs to be worked on.
3) Please specify each variables in Xjf⊗Wjf in section 3. Additionally, please specify what each avertible in the section is and why are they important. Please justify the design choices.
4) Section 4, though has a lot of experimental evaluation but lacks justification and reproducibility. Please explain why such evaluation are chosen and how can be reproduced successfully. Explanation is lacking.

**Summary Of The Paper:**

In this paper, the authors study the components in binary convolution, such as residual connection, Batch Norm, activation function, and structure, for image restoration (IR) tasks. The aim is to conduct systematic analyses to explain each component’s role in binary convolution and discuss the pitfalls of such mechanisms.

**Summary Of The Review:**

1) The section 1 can be rearranged and written better. For example, analysis are provided in numerate bullet points but the paragraphing for each analysis is not consistent. Please rectify that for better readability.
2) It would be better to include some form of content table for better readability at the end of section 1 to describe the progression of the paper texts. Readability of the paper is not good and needs to be worked on.
3) Please specify each variables in Xjf⊗Wjf in section 3. Additionally, please specify what each avertible in the section is and why are they important. Please justify the design choices.
4) Section 4, though has a lot of experimental evaluation but lacks justification and reproducibility. Please explain why such evaluation are chosen and how can be reproduced successfully. Explanation is lacking.

---

> ### Author Response · Authors · 2022-11-16
> **Author feedback to Reviewer WV2K**
>
> **Q1: The section 1 can be rearranged and written better. For example, analysis are provided in numerate bullet points but the paragraphing for each analysis is not consistent. Please rectify that for better readability.**
>
> A1: Thanks for your suggestion! We have adjusted the paragraphing for each analysis in section 1 (see the highlighted texts in section 1 of the revised paper).
>
> ___
>
> **Q2: It would be better to include some form of content table for better readability at the end of section 1 to describe the progression of the paper texts.**
>
> A2: Thanks for your suggestion! We have added the following organization paragraph at the end of section 1.
>
> The remainder of the paper is organized as follows. In Sec. 2, we review related recent literature in IR and BNN. We describe our key observations, analyses, and structure of BBCU in Sec. 3. In Sec. 4, we evaluate the performance of BBCU on various IR tasks (e.g., SR, denoising, and deblocking) and further validate BBCU. We conclude in Sec. 5.
>
> We appreciate further comments and suggestions if there are any. We will take them to improve our paper.
>
> ___
>
>
> **Q3:Please specify each variables in Xjf⊗Wjf in section 3. Additionally, please specify what each avertible in the section is and why are they important. Please justify the design choices.**
>
> A3: Thanks for your suggestion! We have specified each variable in Xjf⊗Wjf in section 3 of the revised paper (see the heightened text in section 3.1).
> We specify each avertible as follows.
>
> (1) BBCUV1: We set an initial BBCU scheme and find that residual connection supplements the full-precision  information loss caused by binarization,  which is important for binarized IR networks to perform accurate pixel prediction. Moreover, different from full-precision IR networks have two Convs in residual convolution, it is best to set residual convolution for each BBCU to preserve more full-precision information (Fig 5 in the paper).
>
> (2) BBCUV2: We try to remove the BN in BBCUV1 to obtain BBCUV2.  BN is not used in full-precision IR networks, which requires accurate pixel prediction and is sensitive to the change of distribution. As we remove the BN in BBCU, we observe that the performance of binarized IR networks would drop, which is caused by the residual misalignment (Fig 2 (b)). The residual misalignment is actually caused by the different Conv forms of residual connection and binary Conv (BC). BC (Eq.1, Fig.7) counts the number of 1 or -1 elements to obtain the results with large values while the value range of full-precision residual connection is close to the input images with the value range of 0 to 1 or -1 to 1.  Overall, the large value range misalignment makes the full-precision information in residual connection covered by BC and the scaling factor of BN can alleviate the value misalignment.
>
> (3) BBCUV3: After we know the real function of BN, we propose a residual alignment (RA) scheme by multiplying the value range of the input image to replace BN. We added RA to the BBCUV2 to obtain BBCUV3. Different from BN narrows the value range of BC, our RA can enlarge the value range of residual connection to realize value range alignment without changing the distribution of image distribution (Fig 2 c, Eqs 7,8). The experiments show that RA can further improve the performance of binarized IR networks. (Tab. 4)
>
> (4) BBCUV4: Based on the above findings, we find that the activation function will damage the negative full-precision information. Thus, we remove the activation function of  BBCUV3 into residual connection to obtain BBCUV4. In this way, the activation function only narrows the negative information of BC.
>
> (5) Arm BBCU on IR networks. We divide IR networks into 4 parts: head, body, upsampling, and tail.  Previous works merely explore the binarization of the body part. Thus, we further develop 4 variants of BBCU to binarize 4 parts (Fig 3).
>
> Section 3 has been modified in the revised paper accordingly.

---

> ### Author Response · Authors · 2022-11-16
> **Author feedback to Reviewer WV2K**
>
> **Q4: Section 4, though has a lot of experimental evaluation but lacks justification and reproducibility. Please explain why such evaluation are chosen and how can be reproduced successfully. Explanation is lacking.**
>
> A4：Thanks for your suggestion!
>
> Justifications for experimental evaluation:
>
> (1) We conduct experiments on various IR tasks to demonstrate that our BBCU is widely effective, and can provide the design guidance of binarized IR networks.
>
> (2) we arm BBCUV1, BBCUV2, BBCUV3, and BBCUV4 on IR networks to show the benefits brought by our improvement (Tab 4).
>
> (3) In Fig 5 and Tab 5, we explore the different properties of residual connection on binarized and full-precision IR networks.
>
> (4) In Fig.6, we explore the amplification factor k of RA. We aim to find the relationship between the optimal amplification factor k and the number of feature channels.
>
> (5) In Tab. 6, we explore the binarization benefit for different parts of IR networks.
>
> Reproducibility:
>
>  Implementation details are described in Sec 4.1. Furthermore, our source code has been included in the supplementary material and we will release it publicly.

---

> ### Author Response · Authors · 2022-11-30
> **Further discussion with Reviewer WV2K**
>
> Dear Reviewer WV2K,
>
> We thank you for the precious review time and valuable comments. Your concerns are mainly about clarity. We have referred to your suggestions and made the following improvements in the revised paper (highlighted texts): (1) We adjusted the paragraphing for each analysis in section 1; (2) We added an organization paragraph at the end of section 1; (3) We have specified each variable in Xjf⊗Wjf and each avertible in section 3 of the revised paper and A3; (4) We further provide more details on justifications for experimental evaluation and reproducibility in section 4 of the revised paper and A4.
>
> We believe our responses have covered your concerns and hope to discuss further with you whether or not your concerns have been addressed.  Please let us know if you still have any unclear parts of our work.
>
> Best,
>
> Authors

---

> ### Author Response · Authors · 2022-12-03
> **Second call for discussion with Reviewer WV2K**
>
> Dear Reviewer WV2K,
>
> We thank you for the precious review time and valuable comments.  We have referred to your suggestions and made the corresponding improvements in the revised paper (highlighted texts), which we believe have covered your concerns.
>
> Due to the approaching deadline of rebuttal and reviewers 2H4s and zzdZ responding that our response can address their concerns, we also hope to discuss further with you whether or not your concerns have been addressed.  Please let us know if you still have any unclear parts of our work.
>
> Best,
>
> Authors

---

> ### Author Response · Authors · 2022-12-10
> **Third call for discussion with Reviewer WV2K**
>
> Dear Reviewer WV2K,
>
> We sincerely thank you for the precious review time and valuable comments. We also appreciate that you acknowledge our motivation and experimental section.
>
> We have referred to your suggestions and revised our paper (highlighted texts). We believe our responses and revisions have covered your concerns.
>
> Up to now, all other reviewers (SdPY, 2H4s, and zzdZ) have responded to our responses that can address their concerns. Please let us know if you still have any unclear parts of our work.
>
> We are looking forward to hearing feedback from you. Thanks for your time!
>
> Best,
>
> Authors

---

### Official Review · Reviewer_SdPY · 2022-10-26

**Confidence:** 4
**Correctness:** 3
**Technical Novelty And Significance:** 2
**Empirical Novelty And Significance:** 3
**Recommendation:** 8

**Clarity, Quality, Novelty And Reproducibility:**

The paper is clear and its results can be reproduced. In terms of novelty, I believe its empirical novelty is more significant.

**Strength And Weaknesses:**

Strengths:

-- The basic binary convolution unit is very simple and effective. IR models equipped with such units outperform existing works.

-- The binary IR models were tested on a wide range of tasks and datasets.

-- The paper is well-motivated, easy to read and well-structured.

Weaknesses:

-- The weight of technical novelty of this work is much lower than that of its empirical novelty. The proposed unit is obtained based on empirical experiments rather than a concrete mathematical foundation.

--  It is not very clear if the improvement comes from RA or RPReLU. What I am curious to see is an ablation study for BBCU-V4 without RA.

-- There is a small degradation in performance when binarizing body and upsampling parts in image super-resolution. I am wondering why it was not explored by prior works. If it was, then please compare.


**Summary Of The Paper:**

This paper presents a method to improve upon binary neural network for image restoration (IR). It shows that batch normalization can help to obtain a good accuracy performance when binarizing IR models. Based on this observation, the authors introduce a basic binary convolution unit that benefits from batch norm properties using residual alignment. It has been shown that IR binary models outperform existing models across different tasks and datasets.


**Summary Of The Review:**

In general, this paper introduces a simple and effective binary unit that further closes the gap between the binary model and its full-precision counterpart. The reasoning and development of the idea is based on empirical results, which its technical novelty is limited.

---

> ### Author Response · Authors · 2022-11-16
> **Author feedback to Reviewer SdPY**
>
> **Q1: The weight of technical novelty of this work is much lower than that of its empirical novelty. The proposed unit is obtained based on empirical experiments rather than a concrete mathematical foundation.**
>
> A1: Yes, the proposed basic binary convolution units were inspired by experimental observations. We discovered the problems of existing BNNs in image restoration through a series of experiments and reasonable analyses. Our extensive experimental comparison and ablation studies can demonstrate the superiority of the proposed method. It would be interesting to justify our findings and methods in a mathematically provable way in the future.
>
> ___
>
>
> **Q2: It is not very clear if the improvement comes from RA or RPReLU. What I am curious to see is an ablation study for BBCU-V4 without RA.**
>
> A2: Thanks for your suggestion! The PSNR (dB) of BBCU-V4 without RA is shown in the following table. We can see that the improvement is mainly from our RA.
>
> | Method       | Set5  | Set14 | B100  | Urban100 | Manga109 |   |   |   |   |
> |--------------|-------|-------|-------|----------|----------|---|---|---|---|
> | BBCUV4       | 31.79 | 28.38 | 27.41 | 25.62    | 29.69    |   |   |   |   |
> | BBCUV4 wo RA | 31.45 | 28.12 | 27.28 | 25.27    | 29.12    |   |   |   |   |
> |              |       |       |       |          |          |   |   |   |   |
>
> ___
>
> **Q3: There is a small degradation in performance when binarizing body and upsampling parts in image super-resolution. I am wondering why it was not explored by prior works. If it was, then please compare.**
>
> A3. Prior works merely investigated binarizing body parts, but did not explore binarizing upsampling parts. Since Convs in the upsampling part have different input and output channels, the proposed residual connection units in SOTA approaches cannot be directly applied in the part. This is a possible reason why previous works ignored the upscaling part.

---

> > ### Comment · Reviewer_SdPY · 2022-12-06
> > **Re: Author feedback to Reviewer SdPY**
> >
> > Thank you for your comments. I am satisfied with response and as such I have increased my recommendation score.

---

> > > ### Author Response · Authors · 2022-12-08
> > > **Thanks Reviewer SdPY for approving our work**
> > >
> > > Dear Reviewer SdPY,
> > >
> > > Thanks for your response. We are happy to see that our response can solve your concerns.
> > >
> > > Best,
> > >
> > > Authors

---

### Author Response · Authors · 2022-11-24
**Response to all reviewers and area chairs for a brief summary**

Dear Reviewers,

We thank all reviewers for their precious review time and valuable comments.  We have responded to each reviewer individually to address any comments. Meanwhile, the manuscript has been updated. We would like to give a brief summary here.

+ We added experiments on BBCU-V4 without RA to further demonstrate the effectiveness of RA.

+ For better readability, we have adjusted the paragraphing for each analysis in section 1 and added the paragraph at the end of section 1 to describe the progression of the paper texts. (highlighted texts in section 1 of the revised paper).

+ For more clarity, we specified each variable in Xjf⊗Wjf and each avertible in section 3. Furthermore, we further introduce justifications for experimental evaluation and reproducibility in A4 to reviewer WV2K and the revised paper.  (highlighted texts in section 3,4 of the revised paper).

+ We added the OPs of compared BBCU in Tab.4 of the revised paper. Besides, we refresh Fig.3 to correct a typo.

+ We added the comparison with other BNN-specific training methods in classification tasks, such as two-stage training, progressive sign function, and knowledge distillation training.

+ We added other compression methods comparisons, such as pruning and knowledge distillation.

+  We deployed our BBCU-M BBCU-L, ReActNet, SRResNet, and SRResNe-Lite on HUAWEI MAIMANG7 to measure real inference speed on Arm devices.

The valuable suggestions of reviewers have greatly benefited our paper. We are happy to see some further discussions to address any of your concerns.

Thanks again.

---

### Author Response · Authors · 2022-12-09
**Updated response to all reviewers and area chairs [Dec 9]**

Dear Reviewers and area chairs,

We thank all reviewers and area chairs for their valuable time and comments. After discussing with reviewers and providing more clarifications/results/analyses, we would like to give a brief response.

All reviewers now agree with the experiments and motivation.

Reviewer SdPY, Reviewer 2H4s, and Reviewer zzdZ now all hold a **positive** side for our work. Our responses have covered their questions. They all now agree with the impressive results on various restoration tasks and the good clarity/writing/organization of our paper.

Moreover, Reviewer 2H4s acknowledges novelty, the potential of our work, reproducibility, and reasonable and extensive experiments, visualizations, and analyses, which can inspire others to investigate more and deeper.

Although Reviewer WV2K gives a negative score in the first round and mainly gives suggestions for clarity, we also made the corresponding improvements in the revised paper (highlighted texts) and call him several times for discussion. However, we do not obtain a response and we believe our revisions and responses have covered his concerns.

We have submitted the demo code of this paper. We would make all the code, trained models, and results available to the public soon. We think that the proposed BBCU implicit is simple, strong, and general and has the potential to be the backbone for binarized IR networks. It is promising to help IR networks applied to resource-limited devices. Moreover, our observations are more important and the identified problems are critical for binarized IR networks and inspire future works. We hope to further investigate this direction in low-level vision together with other researchers.

We thank all reviewers and area chairs again!

Best,

Authors

---

### Decision · Program_Chairs · 2023-01-20

**Decision:**

Accept: poster

**Justification For Why Not Higher Score:**

Level of impact of work and technical novelty.

**Justification For Why Not Lower Score:**

The reviewers overall enjoyed the experimental analysis (as well as myself) and convincing empirical results.

**Metareview: Summary, Strengths And Weaknesses:**

This work looks to understand binary neural networks in the setting of modern CNNs with components such as normalization, skip connection, and various architectural settings. From this analysis, they further propose the basic binary convolution unit (BBCU) so as to outperform previous binary neural networks.

The reviewers overall enjoyed the experimental analysis (as well as myself) and convincing empirical results.

This paper had somewhat high ratings with the exception of one reviewer who gave a score of 3 (reject). Their concerns are effectively two: readability/clarity (which other reviewers didn't have much a problem with) and justification behind chosen experiments (not a significant deal). These are also easy adjustments, and unfortunately, the reviewer did not respond during the discussions or look at the rebuttal. I agree with the majority consensus.

**Note From Pc:**

if the above contains the word "oral" or "spotlight" please see: "oral" presentation means -> notable-top-5% and "spotlight" means -> notable-top-25%. As stated in our emails, we are disassociating presentation type from AC recommendations